# Knowledge, Attitudes and Practices of Flu Vaccination in Hemodialysis Patients

**DOI:** 10.3390/vaccines9020077

**Published:** 2021-01-22

**Authors:** Ada Gawryś, Tomasz Gołębiowski, Dorota Zielińska, Hanna Augustyniak-Bartosik, Magdalena Kuriata-Kordek, Leszek Szenborn, Magdalena Krajewska

**Affiliations:** 1Department of Nephrology and Transplantation Medicine, Wroclaw Medical University, 50-556 Wroclaw, Poland; adagawrys@gmail.com (A.G.); melirus@wp.pl (D.Z.); hanna.augustyniak-bartosik@umed.wroc.pl (H.A.-B.); magdalena.kuriata-kordek@umed.wroc.pl (M.K.-K.); magdalena.krajewska@umed.wroc.pl (M.K.); 2Department of Pediatric Infectious Diseases, Wroclaw Medical University, 50-368 Wrocław, Poland; leszek.szenborn@umed.wroc.pl

**Keywords:** hemodialysis, vaccinations, infections

## Abstract

Background: Hemodialysis (HD) patients have an increased risk of morbidity and mortality due to infections. Despite the positive effect of vaccinations, the implementation of this method of prophylaxis is low. Objectives: This study aimed to explore the knowledge, attitudes and practices of flu vaccination among HD patients of two different dialysis centers. Methods: A total of 193 patients (mean age 63.6 years), who voluntarily agreed to participate in an anonymous survey related to influenza vaccination, were enrolled in this cross-sectional study. Results: A total of 45% of patients declared that they took regular, annual flu vaccination. In this group, 87.4% believed that vaccinations were effective. This opinion strongly correlated with the frequency of regular vaccinations (r = 0.56, *p* < 0.01). Multivariate logistic regression revealed that this opinion is an independent predictor of regular vaccinations with adjusted OR 9.86 (95% CI 4.36, 22.33). Groups of patients who had been irregularly or never vaccinated reject vaccinations for the following reasons: fear of adverse events—29.2%, conviction that vaccination was ineffective—26.4%, and lack of information about vaccination—22.6%. Conclusion: Knowledge among HD patients about the benefits of vaccinations is poor. Therefore, educational activities are required. Active vaccination promotion and education of patients rejecting this method of prevention play a key role in improving standards of care for HD patients.

## 1. Introduction

Infections are one of the main causes of morbidity and mortality in patients with chronic kidney disease (CKD), including those undergoing hemodialysis (HD) or after kidney transplantation [1,2]. CKD patients are at increased risk of infections due to immunoincompetence [3]. At the same time, due to frequent hospitalizations related to hemodialysis procedures, diagnostics and treatment of hemodialysis complications, they also have increased exposure to pathogenic microorganisms [4,5]. Although the immune response to vaccinations is impaired in this population [3], immunization remains an important component of preventive care. Present guidelines related to the care of CKD patients emphasize the importance of vaccination against hepatitis B, influenza virus and pneumococcal infections [6]. Despite the proven positive impact of this method in prevention, the implementation of vaccinations in CKD patients is still too low [4,7,8,9]. Therefore, there is a need to obtain more data on the potential obstacles to vaccination, as well as on the views and opinions on vaccinations in these groups of patients. 

The aim of the study was to investigate the opinion of HD patients about the impact of vaccinations on their health, the degree of implementation of influenza vaccination resulting from this opinion, the reasons for resigning from this form of anti-infective prophylaxis and the sources of information on vaccinations, which are reliable according to the surveyed patients.

## 2. Materials and Methods

The participants were identified from all of the patients undergoing hemodialysis in two dialysis stations (a Clinic Dialysis Center (CDC) and a Satellite Dialysis Center (SDC)). Both dialysis centers are located in Southwest Poland, about 150 km apart, and are mainly inhabited by Caucasian populations with a similar socio-economic structure. Table 1 presents the population characteristics of both dialysis centers, taking into account clinical, demographic and educational data. We did not observe any statistical differences in the above-mentioned indicators. However, these two centers carry out different vaccination practices. The health service in Poland does not reimburse flu vaccinations in patients undergoing hemodialysis. Thus, the burden of purchasing the vaccine rests with the patient, and the general practitioner (GP) is responsible for its implementation (this practice is implemented in CDC). In the SDC, thanks to a grant from the Foundation, the entire flu vaccination procedure, including education, qualification and vaccination, is reimbursed and carried out by the medical staff of the SDC. 

The inclusion criteria for the study were as follows: (1) age 18 years and over, (2) receiving regular dialysis for at least 1 month, (3) ability to provide informed consent, (4) voluntary agreement to participate in an anonymous survey related to the issue of influenza vaccination.

The exclusion criteria included the following: (1) receiving emergency inpatient care within 4 weeks, (2) not willing to participate. Written informed consent was obtained from each patient entering the study. Of the 205 eligible participants, 12 did not agree to take part in the study. 

One hundred ninety-three patients (mean age 63.6 years, (21–86 years), 90 females, with mean dialysis vintage 54.3 months) were enrolled in this cross-sectional study. Demographic and comorbidity data were collected from a direct interview and medical documentation.

Comorbidity included cardiovascular disease (i.e., coronary heart disease and stroke), cancer and diabetes. Coronary heart disease was defined as self-reported complaints or on the basis of a history of myocardial infarction, coronary angioplasty or bypass grafting. Strokes included a history of transient ischemic attack in the past or an ischemic/hemorrhagic event with neurological consequences. Cancers comprised a history of previous neoplastic disease or an active disease. Diabetes patients were those who received insulin or oral antihyperglycemic agents. Combined comorbidity consisted of either coronary heart disease or/and stroke or/and cancer.

The questionnaire included questions about demographic data, comorbidities, and usefulness of vaccinations in maintaining health. Other questions were related to flu vaccination history and vaccination in the current season, as well as the reasons for withdrawing from influenza vaccination. The patients were also asked about sources of information about vaccines they considered reliable.

Statistical analysis was performed using standard software (Statistica Version 13.3, (StatSoft, Tulsa, OK, USA)). Data for continuous variables were expressed as means and standard deviations (±). A paired independent sample t-test was used to compare the means of two continuous variables and chi-square test for categorical variables. The relationship between two variables was evaluated using the Pearson correlation. The results of the univariate and multivariate logistic regression analysis were presented as an odds ratio (OR; 95% confidence intervals (CI)) and adjusted OR (adj. OR; 95% confidence intervals (CI)), respectively. In the multivariable logistic regression model, highly correlated variables were removed to avoid collinearity as described in the Results section. The likelihood ratio (LR) test was used to assess the significance of the entire model. The quality of the model was assessed using the receiver operating characteristic (ROC) with area under curve (AUC) calculation. The completeness of the data was checked by two independent researchers (D.Z., M.K-K.). Therefore, no missing values were observed. Outliers were not removed from the analysis. A *p*-value < 0.05 was considered significant.

The study population was initially divided into two subgroups of patients annually vaccinated against influenza and patients who were either vaccinated irregularly or were not vaccinated against influenza. The baseline characteristics are displayed in Table 1. 

Ethics approval was granted by the Ethics Board of Wroclaw Medical University No KB 732/2017. 

## 3. Results

Eighty-seven patients (45%) declared that they receive regular influenza vaccinations and the remaining 106 patients (54.9%) did not vaccinate regularly or did not vaccinate at all. Both groups were homogeneous in respect of age, sex, education, duration of dialysis treatment, diabetic profile and comorbidity. 

It was shown that patients vaccinated regularly believed more often that vaccinations were effective (87.4% vs. 31.1%, *p* < 0.01) (Table 2). The frequency of regular vaccination strongly correlated with the frequency of the opinion that vaccinations are effective (r = 0.5643, *p* < 0.01).

Development and results of logistic regression model. 

Crude odd ratios (ORs) for all predictors are shown in Table 3. The factors associated with regular vaccinations included the following: patient’s optimistic attitude towards vaccination (vaccines are effective), combined comorbidity, vaccination in the past, higher weight and the vaccination education made by nephrologists. Female sex and reporting concerns were the factors reducing the likelihood of regular vaccinations. 

Based on a correlation analysis between each predictor, the groups of strongly associated predictors were extracted and finally one representative was chosen to avoid collinearity. All the remaining eight predictors (patient’s attitude that vaccines are effective, female sex, combined comorbidity, vaccination concerns, vaccination in the past, information regarding vaccination from the nephrologist, age and weight) were entered into a multivariable logistic regression model to obtain adjusted estimates. The entire model had good quality features with area under curve (AUC) of 0.87 and its significance was confirmed by the likelihood ratio (LR) test (Table 4). Optimistic vaccination attitude (a belief vaccinations are effective) was the strongest, independent predictor for regular vaccination against flu with adjusted OR 9.86 (95% CI 4.36, 22.33). Similarly, increased comorbidity increases the frequency of regular vaccinations (Table 3 and Table 4), but any concerns and obstacles had a negative impact on regular vaccinations (Table 3).

Taking this into account, the study group was divided once more into two groups, in which the first—Group A of patients considered prophylactic vaccinations to be important for maintaining health (109 patients—56%) and the second—Group B of patients considered vaccinations unhelpful or neutral for health (a total of 84 patients—44%) (Table 5). The aim of such a statistical approach was to characterize both groups.

In Group A, there were significantly more patients (76 patients—70%) regularly vaccinated (each year) and patients reported at least once vaccination in the past (74 patients—68%). 

In Group A, only 26.6% (29 patients) reported concerns and obstacles regarding influenza vaccination, while in Group B, the amount was 80.9% (68 patients). The main reasons for abandoning vaccination in both groups were the fear of side effects (9.2% vs. 25%, *p* < 0.05) and lack of knowledge about the need to vaccinate annually (8.3% vs. 17.9%, *p* = 0.08).

In both groups (A and B), nephrologists were the most reliable source of information about vaccinations, generally indicated by 76% of the respondents (Group A 81.7% vs. Group B 67.9%). Family doctors were also indicated as an important source of knowledge about vaccinations (Group A 44% vs. Group B 57.1%). Only a few people mentioned the press (medical, popular, internet) as well as friends and family.

In the Satellite Dialysis Station, the group of patients who declared having vaccinated against flu at least once in their lifetime constitutes 71 out of 90 patients (78.8%) and included 67 patients (74%) vaccinated annually. Among patients from the Clinic Dialysis Station, 25 out of 103 patients (24.3%) declared having vaccinated against flu. This group included 20 patients (19%) vaccinated annually. 

## 4. Discussion

In our recent study, we analyzed vaccination practices in two dialysis centers. These centers differed in the method of flu prevention. The results clearly show that the number of vaccinated patients depends on the patient’s belief that vaccination is beneficial and effective. This belief increases the chance of vaccination by about nine times (OR 9.86 (95% CI 4.36, 22.33)). Nephrologists educating the patient play a significant role (Table 2). Additionally, an increased comorbidity significantly increased the frequency of regular vaccinations (Table 4). This was not a surprise to us because patients with previous diseases more easily follow the suggestions of the attending physician regarding the treatment plan.

Epidemiological studies indicate that pneumonia and sepsis are the most common infectious complications in the CKD patient population [10,11,12]. There is strong evidence that in the general population vaccinations are effective methods of preventing infections [13,14]. There are data showing the beneficial effects of immunization in CKD patients, which is of particular importance considering the fact that hospitalizations for bacteremia or sepsis occur four times more often in this group compared to those without kidney diseases [15,16,17].

Among patients who received HD, vaccination against influenza A and B is associated with a reduced risk of infection-related hospitalizations, hospitalizations for influenza or pneumonia, all-cause mortality, infection-related mortality, and cardiac-related mortality [13,18]. Influenza, hepatitis B, and pneumococcal vaccines are now widely recommended for CKD patients in most countries by local, regional, or national immunization advisory boards. Current recommendations of KIDIGO 2012 encourage annual flu vaccination, vaccination against Streptococcal pneumonia and HBV vaccination in chronic kidney disease patients, especially those who are likely to require renal replacement therapy (glomerular filtration rate GFR <15 mL/min/1.73 m²) [6]. Despite the proven positive effect of influenza vaccination among dialysis patients, including lower probability of viral pneumonia, respiratory failure, hospital stay and lower mortality [19,20], a large number of patients are not aware of the benefits of vaccinations. This translates into the number of vaccinated patients.

The present study revealed important findings regarding vaccination of HD patients. Firstly, the frequency of regular vaccinations strongly correlated with the frequency of the opinion that vaccinations are effective. Most patients (87%) who do not believe in the positive effects of vaccinations or have no knowledge about them did not uptake the annual flu vaccination. Similar findings are presented in Bertolodo et al.’s study from southern Italy [21]. In a group of 700 adults with chronic diseases, 64.7% of the participants were aware that influenza can be prevented with vaccines and that patients with chronic diseases are at higher risk of developing severe complications. This was the reason that 42.1% received influenza vaccine in the last season, and 46.9% declared the will to receive influenza vaccination in the next season [21]. Having a positive attitude towards the usefulness of influenza vaccination was found to be significantly associated with vaccine uptake among higher risk patients in similar studies in the US and France [22,23].

The alarming fact is that the percentage of chronic ill patients regularly vaccinated against influenza is still low. In our study, 45% of HD patients declared receiving regular influenza vaccinations. Therefore, vaccination coverage remains far below the recommended target of 75% [24]. According to the WHO data from European Region on influenza vaccination coverage, 33 (72%) countries recommended influenza vaccinations for older people and people with chronic diseases in 2014/2015, and the reported vaccination uptake ranged from 0.03% to 76.3%, with a median of 34.4% [24]. Among 44 countries with influenza vaccine recommendations for persons with specific chronic illnesses (pulmonary, renal, hepatic, neurologic, and immunosuppressive diseases including HIV infection) in 2014/2015, fourteen (32%) provided information on coverage and most countries reported rates below 40% [24]. Lower rates have been reported in studies conducted, for example, in Ireland (29.1%) [25] and Germany (23%) [26]. 

A fairly favorable situation among HD patients is observed in the area of hepatitis B vaccination. Vigilant compliance with the guidelines and regular virology testing are the cornerstones of effective control of chronic hepatitis in the setting of HD [17,27,28]. Thanks to such a strategy of patient protection, patients’ knowledge about own viral status is associated with high awareness and understanding the need for vaccinations. 

As reported in other studies [29,30], many patients are concerned with vaccine safety. The myths about vaccines and the lack of education about flu vaccine in particular encourage patients to believe that the vaccination is ineffective. Furthermore, as shown in our study, 29.2% of unvaccinated responders indicate side effect concerns as the main reason for vaccination abandonment (Table 2). 

Influenza vaccination is generally well tolerated. In the placebo-controlled studies performed in adults, the most common vaccine event following injection of trivalent inactivated influenza vaccines (TIVs) was pain, redness and swelling at the injection site for less than 2 days after vaccination. Other events included fever, headache, muscle pain, and malaise, often referred to as flu-like symptoms. Serious allergic and anaphylactic reactions occurred in response to various components of the flu vaccine but were very rare and observed in 0.65/1,000,000 of vaccinated patients [31]. The educational role of healthcare professionals about the actual vaccine reactions, true side effects and the benefits related to regular vaccination is also important. Well-educated patients from our study who received vaccines annually declared no fear of side effects.

Not only should the benefits of vaccinations be shown, but also the possible health risks of rejecting vaccinations. The low risk of post-vaccination reactions and multiplied economic costs in the event of a flu illness (purchase of drugs, absenteeism, forced hospitalization) should constitute an argument encouraging vaccinations. The benefits of vaccinations are extremely important in high-risk groups, not only in respect to the flu infection itself. A study from Taiwan showed the beneficial effects of influenza vaccination in reducing the risk of stroke. These studies have shown that even a single flu vaccination significantly reduces the risk of hospitalization due to ischemic stroke, and vaccination in one season reduced the risk of stroke by up to 24% [32]. Similar results have been confirmed by other medical centers, where it has been observed that vaccinations during the epidemic season reduce the risk of ischemic stroke [33,34,35]. Another argument in favor of influenza vaccination is the positive impact on cardiovascular incidents. There was an association between influenza vaccination and the reduction in myocardial infarction risk in patients 50 years and older [36]. In Siriwardena et al.’s study, influenza vaccination was associated with a 19% reduction in the rate of acute myocardial infarction [36].

The next important finding derived from our study is that nephrologists were most frequently indicated as a source of information about vaccinations (in 76% of all patients—146 out of 193) (Table 2). This is probably due to the fact that, due to the dialysis program (usually three times every week), contact is more accessible and more frequent than with a GP. Between 68% and 82% of HD patients indicate a nephrologist as a reliable person with regard to vaccination counseling. Advice given by health care providers may be an important key predictor of influenza vaccination. Olatunbous et al. emphasized in a cross-sectional survey that the advice from doctors about the importance of vaccinations strongly influenced the decision to get vaccinated. In this study, 97.6% of diabetic patients vaccinated in Pretoria were encouraged to do so by their doctors [37]. In addition, in Tan et al.’s study, 241 (78.5%) participants (diabetic patients) indicated that the advice given to them by healthcare professionals was the main guiding factor for getting vaccinated, but almost 65% had never been advised on influenza vaccination [38].

All of these observations are confirmed in the group of HD patients from Satellite Dialysis Center (SDC). Thanks to the Foundation’s cooperation, the SDC offers HD patients free flu vaccinations. Additionally, vaccinations are carried out at the SDC. All patients have the possibility to learn about flu immunization, discuss concerns and fears of side effects with the medical staff of the HD unit. Issues related to vaccination costs were probably the most important, as patients on dialysis live off their pension and benefits. Additionally, the time required for conducting vaccinations in a GP practice along with the need to register for vaccinations to the GP and be qualified for vaccinations make the approach in a HD center comfortable. Promotion of vaccinations by nephrologists as well as payment and organization of vaccinations at the HD unit during “dialysis visits” contribute significantly to a high degree of flu vaccination (74%) among SDS patients. In comparison, at the Clinic Dialysis Center (CDC), where influenza vaccines are not offered free of charge, only 25 from 103 patients (24.3%) declare getting vaccinated and 20 patients (19%) annually.

Finally, it is worth mentioning that, in the presented study, the annual influenza vaccination coverage in HD patients was 45% and was higher than in the general Polish population, which has been below 5% in recent years, approximately 3.9% in 2018/2019, 4.12% in 2019/2020, and is currently one of the lowest in Europe [39].

Our results are in agreement with those obtained by other researchers. Dower et al. have found that influenza vaccination coverage in adults with chronic diseases was 47%. The study was conducted on 2203 adults with asthma, diabetes or a cardiovascular condition [40]. 

In light of the ongoing outbreak of coronavirus disease 2019 (COVID-19), routine immunization with available vaccines and observation of management rules have become increasingly important. As the symptoms and signs of seasonal flu and flu-like illnesses could be similar to COVID-19, counseling regarding routine influenza vaccination should be provided to all patients. The role of telehealth or home vaccinations by healthcare personnel might be applied in order to vaccinate all unvaccinated patients.

A few limitations should be underlined when assessing our findings. Owing to the cross-sectional study design, we were unable to comment on the dynamic nature of patients’ opinions in regard to vaccination. As the study used an anonymous questionnaire, there might be a subjective element in understanding and answering the questions included in the survey. Our data are also limited by the fact that we recruited dialysis patients from only two in-center HD units. Different vaccination practices in the groups of patients treated in both centers (CDC and SDC) are the reason why all our data should not be extrapolated to the entire dialysis population in Poland.

## 5. Conclusions

Our results indicate that the lack of knowledge, organization and refunds are the main barriers to vaccination. Patients who are aware of the positive impact on vaccination uptake it more often. That is why the advice of health care providers may be an important key predictor of influenza vaccination. Medical personnel should continue to educate and encourage all patients to get vaccinated for influenza. Active promotion of vaccinations in HD patients provided by nephrologists plays a key role in raising the standards of care for dialysis patients.

## Figures and Tables

**Table 1 vaccines-09-00077-t001:** Population characteristics in CDC and SDC.

Variables	CDC (No = 103)	SDC (No = 90)	*p*
Age y. ± SD	64.48 ± 14.64	62.54 ± 14.11	0.354
Female (1) No (%)	53 (51)	37 (41)	0.385 *
Basic education (1) No (%)	30 (29)	30 (33)	0.648 *
High school education (1) No (%)	51 (50)	53 (59)	0.476 *
Higher education (1) No (%)	22 (21)	9 (10)	0.067 *
Weight kg ± SD	75.41 ± 18.11	80.27 ± 16.87	0.059
BMI kg/m² ± SD	27.42 ± 5.78	28.37 ± 6.19	0.271
Diabetes (1) No (%)	30 (29)	37 (41)	0.226 *
Combined comorbidity (1) No (%)	32 (31)	46 (51)	0.066 *

* Chi-square test. Abbreviations: CDC, Clinic Dialysis Center; SDC, Satellite Dialysis Center; BMI, body mass index.

**Table 2 vaccines-09-00077-t002:** Baseline characteristics of the study groups.

Variables	Patients Regularly Vaccinated against Influenza—One a Year (No = 87)	Patients Irregularly or Not Vaccinated against Influenza (No = 106)	*p*
Age y. ± SD	64.09 ± 14.06	62.94 ± 14.83	0.601
Female No (%)	31 (30.1)	59 (55.7)	0.09 *
Basic education No (%)	32(36.8)	28(26.4)	0.31 *
High school education No (%)	46 (52.9)	58 (54.7)	0.89 *
Higher education No (%)	9 (10.3)	20 (18.9)	0.15 *
Weight kg ± SD	75.03 ± 18.88	82.66 ± 21.03	0.291
Diabetes No (%)	32 (36.8)	35 (33)	0.70 *
Combined comorbidity No (%)	45 (51.7)	34 (32)	0.08 *
HD vintage m ± SD	57.96 ± 53.50	49.72 ± 56.85	0.550
Patients from SDC No (%)	67 (77)	23 (21.7)	<0.01 *
**Vaccination concerns and obstacles**			
Patient did not know that they should be vaccinated against influenza No (%)	0 (0)	24 (22.6)	<0.01 *
Patient was afraid of vaccine side effects No (%)	0 (0)	31 (29.2)	<0.01 *
Patient thinks that vaccination takes too much time No (%)	0 (0)	4 (3.8)	0.07 *
Patient thinks that the vaccine is too expensive No (%)	0 (0)	6 (5.7)	<0.05 *
Patient thinks that influenza vaccinations are ineffective No (%)	1 (1.1)	28 (26.4)	<0.01 *
Patients with any concerns and obstacles No (%)	1 (1.1)	96 (90.6)	<0.01 *
**Who or what is a reliable source of information for patient about vaccinations?**			
Nephrologist No (%)	72 (82.8)	74 (70)	0.44 *
Family doctor No (%)	36 (41.4)	60 (56.6)	0.22 *
Medical press No (%)	2 (2.2)	14 (13.2)	<0.05 *
Daily or popular press No (%)	0 (0)	6 (5.7)	<0.05 *
Family or friends No (%)	2 (2.2)	6 (5.7)	0.26 *
Vaccination websites No (%)	0 (0)	6 (5.7)	<0.05 *
Patients who believe vaccination is effective No (%)	76 (87.4)	33 (31.1)	<0.01 *
Patients vaccinated in the past No (%)	65 (74.7)	30 (28.3)	<0.01 *

* Chi-square test. Abbreviations: SDC, Satellite Dialysis Center; SD, standard deviation.

**Table 3 vaccines-09-00077-t003:** Univariate logistic regression. Variables correlated to vaccination status (patients regularly, annually vaccinated against influenza).

Variables	OR	95% CI	*p*	Estimate
Age (y.)	0.994	0.975–1.014	0.579	−0.006
Female (1)	0.441	0.246–0.790	0.006	−0.819
Basic education (1)	1.621	0.878–2.993	0.123	0.483
High school education (1)	0.929	0.526–1.640	0.798	−0.074
Higher education (1)	0.496	0.213–1.154	0.104	−0.701
Weight (kg)	1.024	1.008–1.040	0.004	0.023
Diabetes (1)	1.180	0.651–2.139	0.585	0.166
Combined comorbidity (1)	2.269	1.263–4.076	0.006	0.819
HD vintage (m)	0.997	0.992–1.003	0.304	−0.003
**Vaccination concerns and obstacles**				
Patient did not know that they should be vaccinated against influenza (1)	0.000	0.000–0.244	0.997	−20.262
Patient was afraid of vaccine side effects (1)	0.000	0.000	0.996	−20.351
Patient thinks that vaccination takes too much time (1)	0.000	0.000	0.997	−18.044
Patient thinks that the vaccine is too expensive (1)	0.000	0.000	0.997	−19.064
Patient thinks that influenza vaccinations are ineffective (1)	0.032	0.004–0.244	0.001	−3.430
Patients with any concerns and obstacles (1)	0.001	0.000–0.010	0.000	−6.716
**Who or what is a reliable source of information for patient about vaccinations?**				
Nephrologist (1)	2.076	1.037–4.154	0.0391	0.730
Family doctor (1)	0.541	0.305–0.961	0.0361	−0.614
Medical press (1)	0.155	0.034	0.015	−1.867
Daily or popular press (1)	0.000	0.000	0.997	−19.064
Family or friends (1)	0.392	0.077–1.994	0.259	−0.936
Vaccination websites (1)	0.000	0.000	0.997	−19.064
Patients who believe vaccination effective (1)	15.284	7.189–32.492	0.000	2.727
Patients vaccinated in the past (1)	7.485	3.938–14.225	0.000	2.013

Abbreviations: OR, odds ratio; CI, confidence intervals.

**Table 4 vaccines-09-00077-t004:** Multivariate logistic regression.

Variables	adj. OR	95% CI	*p*	Estimate	LR * (*p*)
Patients who believe vaccination are effective (1)	9.863	4.357–22.326	0.000	2.289	<0.01
Vaccinated in the past (1)	5.179	2.367–11.332	0.000	1.645	<0.01
Combined comorbidity (1)	2.448	1.127–5.318	0.024	0.895	<0.01
Female (1)	0.408	0.189–0.882	0.023	−0.896	<0.05

Abbreviations: adj. OR, adjusted odds ratio; CI, confidence intervals, LR, * likelihood ratio (LR) test.

**Table 5 vaccines-09-00077-t005:** Comparison of two groups of subjects (patients who believe vaccination is effective (Group A) and patients who believe vaccination is ineffective (group B).

Variables	Group A Vaccination Effective (No = 109)	Group B Vaccination Ineffective (No = 84)	*p*
Age y. ± SD	63.1 ± 14.6	64.2 ± 14.2	0.61
Female No (%)	45 (41.2)	45 (53.5)	0.308 *
Basic education No (%)	33 (30.5)	27 (32.1)	0.840 *
High school education No (%)	64 (58.7)	40 (47.6)	0.398 *
Higher education No (%)	12 (11)	17 (20.3)	0.127 *
Weight kg ±SD	80.1 ± 21	76.3 ± 18.9	0.19
Diabetes No (%)	40 (36.7)	25 (29.8)	0.47 *
Combined comorbidity No (%)	79 (72)	53 (63)	0.55 *
HD vintage m ± SD	56.2 ± 61.2	51.7 ± 46.1	0.58
Patients from SDC No (%)	69 (63.3)	21 (25)	<0.01 *
**Vaccination concerns and obstacles**			
Patient did not know that they should be vaccinated against influenza No (%)	9 (8.3)	15 (17.9)	0.08 *
Patient was afraid of vaccine side effects No (%)	10 (9.2)	21 (25)	<0.05 *
Patient thinks that vaccination takes too much time No (%)	1 (0.9)	3 (3.6)	0.21 *
Patient thinks that the vaccine is too expensive No (%)	2 (1.8)	4 (4.8)	0.26 *
Patients with any concerns and obstacles No (%)	29 (26.6)	68 (8.9)	<0.01 *
**Who or what is a reliable source of information for patient about vaccinations?**			
Nephrologist No (%)	89 (81.7)	57 (67.9)	0.41 *
Family doctor No (%)	48 (44)	48 (57.1)	0.29 *
Medical press No (%)	7 (6.4)	9 (10.7)	0.32 *
Daily or popular press No (%)	2 (1.8)	4 (4.8)	0.26 *
Family or friends No (%)	2 (1.8)	6 (7.1)	0.08 *
Vaccination websites No (%)	1 (0.9)	5 (6)	0.053 *
Patients regularly vaccinated against influenza—one a year No (%)	76 (70)	11 (13)	<0.01 *
Patients vaccinated in the past No (%)	74 (67.9)	21 (25)	<0.01 *

* Chi-square test. Abbreviation; SDS, Satellite Dialysis Station.

## Data Availability

Not applicable.

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
