# Peer review of "Knowledge, Attitudes and Practices of Flu Vaccination in Hemodialysis Patients"

_vaccines, 2021, doi:10.3390/vaccines9020077_

Round 1
Reviewer 1 Report
Overall this well-designed study is based on epidemiological data.
My suggestions to the authors
1) Is the informed consent was taken from the subjects included in this study? Add this into the method section.
2) study included 193 patients . Authors can compare two risk commodities in the individuals who were having two risk factors during the vaccination. I think it's a small number but gives a good indication.
Author Response
Reviewer 1
Overall this well-designed study is based on epidemiological data.
My suggestions to the authors
1) Is the informed consent was taken from the subjects included in this study? Add this into the method section.
Response: Thank you for this suggestion. We have added information on informed consent in the Methods section.
2) study included 193 patients . Authors can compare two risk commodities in the individuals who were having two risk factors during the vaccination. I think it's a small number but gives a good indication.
Response: In the study we assessed the influence of various factors on the regular, annually vaccination in group of hemodialysis patients. Indeed, we agree that increasing the number of observations could have a positive effect on the statistical significance. The sample size for logistic regression was based on the following work: Peduzzi et al . A simulation study of the number of events per variable in logistic regression analysis. DOI: https://doi.org/10.1016/S0895-4356(96)00236-3 and was performed according following equations:
N = 10*k / p
In previous study (Dower, J.; Donald, M.; Begum, N.; Vlack, S.; Ozolins, I. Patterns and determinants of influenza and pneumococcal immunisation among adults with chronic disease living in Queensland, Australia. Vaccine 2011, 29,3031–3037) regular, annually vaccination was 47%, so proportion of positive cases was p=0,45. We used 8 predictors (k) in analysis.
N=10*8/0,47, so
The minimal number of observations was N=170.
Moreover, the model has achieved statistical significance in the likelihood ratio (LR) test.
The following information was included in Results section
The entire model had good quality features with area under curve (AUC) of 0.87 and its significance was confirmed by the likelihood ratio (LR) test (Table 4).
Reviewer 2 Report
Authors performed analysis of vaccination practices in patients undergoing dialysis in two dialysis centers and identified that ~45% of patients undergo regular flu vaccinations. They investigated potential causes for limited prevalence of vaccination and conclude that the lack of patient education and vaccination refunds pose barriers to vaccination.
Comments:
1) Methods would benefit from clarifications:
- It is unclear where the testing centers are located (same or different countries or regions within the country which could be relevant due to potential regional differences in the population composition and behavior)
- Description of statistical methods should be added to methodology, including which test(s) were used, which cutoffs and multiple testing corrections were used, how missing values (if any) were handled, how were potential outliers and confounding factors (such as highly correlated variables) handled and other information as appropriate
2) Tables should be re-formatted for clarity: as it is, tables 1,2 and 4 quite difficult to read, they would benefit form re-formatting
3) Study would benefit form comparison of populations from two centers: Study examines participants from two centers which use different vaccination practices. As such, it is important to compare these groups to identify if centers have different vaccination rates and patient factors (such as age, sex, BMI and education levels) to avoid potential bias in the analysis where the results are driven by one of the centers. In addition, such comparison would add to the value of the study as it might reveal if potential differences in vaccination rates between centers are driven by differences in group compositions and vaccination practices.
Author Response
Reviewer 2
Authors performed analysis of vaccination practices in patients undergoing dialysis in two dialysis centers and identified that ~45% of patients undergo regular flu vaccinations. They investigated potential causes for limited prevalence of vaccination and conclude that the lack of patient education and vaccination refunds pose barriers to vaccination.
Comments:
1) Methods would benefit from clarifications:
- It is unclear where the testing centers are located (same or different countries or regions within the country which could be relevant due to potential regional differences in the population composition and behavior)
Response: We appreciate this suggestion. We made such an analysis. However, no differences were found in socio-economic, demographic and clinical indicators. We have added the following information in the methods section:
Both dialysis centers are located in south-west Poland, about 150 km apart, and are mainly inhabited by Caucasian population with a similar socio-economic structure. Table 1 presents the population characteristics of both dialysis centers, taking into account clinical, demographic and educational data. We did not observe any statistical differences in the above-mentioned indicators.
Description of statistical methods should be added to methodology, including which test(s) were used, which cutoffs and multiple testing corrections were used, how missing values (if any) were handled, how were potential outliers and confounding factors (such as highly correlated variables) handled and other information as appropriate
Response: We have added following description of statistical methods in Methods section.
Statistical analysis was performed using standard software (Statistica Version 13.3, (StatSoft, Tulsa, OK, USA)). Data for continuous variables were expressed as means and standard deviations (±). Paired independent sample t-test was used to compare the means of two continuous variables and chi-square test for categorical variables. The relationship between two variables was evaluated using the Pearson correlation. The results of univariate and multivariate logistic regression analysis were presented as an odds ratio (OR; 95% confidence intervals (CI)) and adjusted OR (adj. OR; 95% confidence intervals (CI)), respectively. In the multivariable logistic regression model highly correlated variables were removed to avoid collinearity as described in Results section. The likelihood ratio (LR) test was used to assess the significance of the entire model. The quality of the model was assessed using Receiver Operating Characteristic (ROC) with area under curve (AUC) calculation. The completeness of the data was checked by two independent researchers (D.Z., M.K-K.). Therefore no missing values were observed. Outliers were not removed from the analysis. A p-value < 0.05 was considered significant.
No cut-off values were assessed and multiple testing corrections were not used in the study.
2) Tables should be re-formatted for clarity: as it is, tables 1,2 and 4 quite difficult to read, they would benefit form re-formatting
Response: We apologize for the table format, which was probably due to changes during the editing process. The tables were accordingly re-formatted.
3) Study would benefit form comparison of populations from two centers: Study examines participants from two centers which use different vaccination practices. As such, it is important to compare these groups to identify if centers have different vaccination rates and patient factors (such as age, sex, BMI and education levels) to avoid potential bias in the analysis where the results are driven by one of the centers. In addition, such comparison would add to the value of the study as it might reveal if potential differences in vaccination rates between centers are driven by differences in group compositions and vaccination practices.
Response: Ones more time we thank you for this remark. Indeed, the comparison of dialysis stations population in both centers is important, therefore demographic, clinical and educational level analysis was performed. The results are presented in new Table 1. As mentioned above, the analysis did not reveal any significant differences. Taking this into account, the Discussion section was supplemented with this information as well as the fact that both centers were different in terms of vaccination practice.
In our recent study, we analyzed vaccination practices in two dialysis centers. These centers differed in the method of flu prevention. The results clearly show that the number of vaccinated patients depends on the patient's belief that vaccination is beneficial and effective. This belief increases the chance of vaccination by about 9 times (OR 9.86 (95% CI 4.36, 22.33)). Nephrologists educating the patient play a significant role (Table 2). Additionally, an increased comorbidity significantly increased the frequency of regular vaccinations (Table 4). This was not a surprise to us because patients with previous diseases easier follow the suggestions of the attending physician regarding the treatment plan.
and in Limitation part of the Discussion
Different vaccination practices in the groups of patients treated in both centers (CDC and SDC) are the reason why all our data should not be extrapolated to the entire dialysis population in Poland.
The order of the other tables has been changed accordingly.
Additionally, we changed incorrectly entered data in the text - all changes are marked in blue.